# Structural and Hemodynamic Changes of the Right Ventricle in PH-HFpEF

**DOI:** 10.3390/ijms23094554

**Published:** 2022-04-20

**Authors:** Maria Barilli, Maria Cristina Tavera, Serafina Valente, Alberto Palazzuoli

**Affiliations:** 1Department of Medical Biotechnologies, Division of Cardiology, Le Scotte Hospital, 53100 Siena, Italy; 2Cardio-Thoracic and Vascular Department, Division of Cardiology, Le Scotte Hospital, 53100 Siena, Italy; cristina.tavera@ao-siena.toscana.it (M.C.T.); serafina.valente@ao-siena.toscana.it (S.V.); 3Cardiovascular Diseases Unit, Cardio-Thoracic and Vascular Department, Le Scotte Hospital, 53100 Siena, Italy

**Keywords:** HFpEF, pulmonary hypertension, right ventricle, imaging

## Abstract

One of the most important diagnostic challenges in clinical practice is the distinction between pulmonary hypertension (PH) due to primitive pulmonary arterial hypertension (PAH) and PH due to left heart diseases. Both conditions share some common characteristics and pathophysiological pathways, making the two processes similar in several aspects. Their diagnostic differentiation is based on hemodynamic data on right heart catheterization, cardiac structural modifications, and therapeutic response. More specifically, PH secondary to heart failure with preserved ejection fraction (HFpEF) shares features with type 1 PH (PAH), especially when the combined pre- and post-capillary form (CpcPH) takes place in advanced stages of the disease. Right ventricular (RV) dysfunction is a common consequence related to worse prognosis and lower survival. This condition has recently been identified with a new classification based on clinical signs and progression markers. The role and prevalence of PH and RV dysfunction in HFpEF remain poorly identified, with wide variability in the literature reported from the largest clinical trials. Different parenchymal and vascular alterations affect the two diseases. Capillaries and arteriole vasoconstriction, vascular obliteration, and pulmonary blood fluid redistribution from the basal to the apical district are typical manifestations of type 1 PH. Conversely, PH related to HFpEF is primarily due to an increase of venules/capillaries parietal fibrosis, extracellular matrix deposition, and myocyte hypertrophy with a secondary “arteriolarization” of the vessels. Since the development of structural changes and the therapeutic target substantially differ, a better understanding of pathobiological processes underneath PH-HFpEF, and the identification of potential maladaptive RV mechanisms with an appropriate diagnostic tool, become mandatory in order to distinguish and manage these two similar forms of pulmonary hypertension.

## 1. Introduction

Pulmonary hypertension (PH) is a progressive disorder of the pulmonary vasculature that has received great attention over the past years, with significant advances in the comprehension of pathophysiology, hemodynamic mechanisms, and the involvement of the right side of the heart. The latest European guidelines define PH as a “hemodynamic and pathophysiological condition characterized by an increase in mean pulmonary arterial pressure (PAPm) of up to ≥25 mmHg at rest, assessed by right heart catheterization” [1]. The 6th World Symposia on Pulmonary Hypertension (WSPH), held in February 2018, proposed to change the hemodynamic definition of PH by reducing the threshold for abnormal PAPm to >20 mmHg, as it already represents a value above the 97.5th percentile. Additional outcome studies are needed to define the most accurate cut-off and evaluate the efficacy of PAH medications in patients with mPAP spanning from 21 to 24 mmHg [2].

Left heart diseases (LHDs), or group 2 PH, are recognized as the most common causes of PH [1]. Their underlying mechanism has been studied since the 1930s [3], with early research on mitral stenosis. Long-standing increases of left-side filling pressure generate chronic elevation in pulmonary venous pressure and subsequent pulmonary arterial vasoconstriction and remodelling [2,4,5,6]. To date, the hemodynamic definition of PH-LHD is post-capillary, with a PAPm of ≥25 mmHg and a pulmonary arterial wedge pressure (PAWP) of >15 mmHg. The post-capillary PH is subclassified into isolated post-capillary PH (IpcPH) when the diastolic pulmonary gradient (DPG) is <7 mmHg and/or the pulmonary vascular resistance (PVR) is ≤3 Wood Units (WU), and into combined post-capillary PH (CpcPH) when the DPG is ≥7 mmHg and/or the PVR is >3 WU [2]. This last hemodynamic phenotype mostly affects patients with heart failure with preserved ejection fraction (HFpEF), representing the most complex situation, as it may be misdiagnosed with group 1 PH or pulmonary arterial hypertension (PAH) [7]. Given its positive response to therapy in PAH, the role of PH in HFpEF is currently under intense scrutiny [5], as it accounts for about 50% of HF patients, with a large epidemiological impact and a lack of effective management [7].

## 2. Pulmonary Hypertension: PAH and PH-HFpEF

In the context of heart failure with preserved ejection fraction, there is increasing evidence of an overlap between the advanced stages of PH-HFpEF and those of the PAH phenotype. Most PH-HFpEF patients have isolated post-capillary PH (IpcPH) [8], where a sustained PAWP elevation leads to pulmonary venous vascular remodelling without causing an increased PVR, which remains at ≤3 WU [9]. Thus, the typical hemodynamic profile of PH-LHD patients is characterized by a PAPm that is not excessively elevated (25–40 mmHg) by an increased PAWP (usually >20 mmHg), a low cardiac index with an elevated transpulmonary gradient (TPG > 12 mmHg), or consistently elevated right atrial pressure (>10 mmHg) [7]. Conversely, PAH disorder is rather defined by a PAPm of ≥25 mmHg at rest, a pulmonary arterial wedge pressure (PAWP) of ≤15 mmHg, and pulmonary vascular resistance (PVR) of >3 Wood Units (WU), within the context of normal LA pressures [1,2]. Moreover, the disease process in PAH appears to be located primarily in the arterioles, without the involvement of venules [5,9,10,11]. When PH-HFpEF progresses, the pathology begins to affect distal pulmonary arteries, causing the development of a pre-capillary component of PH—that is, an elevated PVR (>3WU)—and the constellation of combined pre- and post-capillary PH (CpcPH) [9,12].

The identification of a pre-capillary component in PH-HFpEF is crucial, as CpcPH has a considerably worse prognosis with a reduced exercise capacity, different management, and a different response to PAH-specific vasodilator compared to IpcPH [7,13,14,15]. It has been proven, in a retrospective analysis of 2587 patients, that markers of CpcHF, such as a DPG of ≥12 mmHg, TPG of ≥12 mmHg, and PVR of ≥3 WU, were predictors of mortality and heart failure hospitalisations [16].

The 5th WSPH proposed a 3-step approach in order to help differentiate group 2 and PAH using pre-test probability scores to define patients who would more likely undergo the invasive hemodynamic definition and to predict the outcome and prognosis [17]. Compared to the PAH phenotype, PH-HFpEF is characterized by specific demographic, clinical, and echocardiographic features, including older age, female gender, hypertension, higher rate of atrial fibrillation, and metabolic syndrome (hyperlipidaemia, obesity, diabetes, and hypertension) [18] (Table 1). Many scores have been proposed so far [18,19,20,21,22,23,24], integrating features analysed from retrospective single centre studies; some have even suggested that simple clinical characteristics alone, without echocardiographic or hemodynamic data, were able to differentiate PH-HFpEF from PAH [25]. No score has been externally validated so far, but a similar approach has recently been proposed to assess HFpEF among patients with chronic heart failure. The research for a CpcPH-HFpEF and PAH phenotype has led to the identification of a sub-classification for patients in “typical PAH”, “atypical PAH”, and PH-HFpEF (CpcPH) [7,26]. The second category includes older patients with PAH and a high burden of cardiovascular comorbidities (defined by the presence of at least three risk factors for left ventricular dysfunction, including the ones listed above). The phenotypes were first identified in the AMBITION (Ambrisentan and Tadalafil in Patients with Pulmonary Arterial Hypertension) trial, where patients with “atypical PAH” were excluded from the primary analysis set [27]. An analysis of the 5935 patients from the multinational Comparative, Prospective Registry of Newly Initiated Therapies for Pulmonary Hypertension (COMPERA) with PH and treated with at least one targeted PH therapy showed how the effects were less pronounced in patients with CpcPH-HFpEF in comparison with “typical” PAH patients, with the “atypical” category standing in between [28]. This supports the hypothesis that there is a continuum from “typical” PAH and HFpEF combined PH, but more studies are needed in order to evaluate the effect of PAH treatments on CpcPH, as their use is still not recommended by the task force in any kind of PH-LHD [7].

## 3. Epidemiology

Data on PH prevalence and severity are often unclear, with a wide range of inclusion criteria [1,25] due to a non-uniform application of RHC diagnostic standards [5] and a lack of standard randomization selection with observations obtained in different stages of the disease [4]. PAH prevalence and incidence in Europe are approximately around 15–60 subjects per million population and 5–10 cases per million per year, respectively [29,30,31]. PH-HFpEF has a higher impact but with a wider variability of data; the reported prevalence among the overall HFpEF population ranges from 18% to 83%, depending on the group studied [32]. Notably, the HFpEF definition criteria have been changed during the last decades, and the most commonly accepted inclusion criteria are mainly based on the EF threshold and natriuretic peptide increase. Most trials did not consider a detailed non-invasive doppler analysis or the left ventricular and left atrial geometry measurement. Therefore, selection biases are based on data mostly concerning the echocardiographic estimation of systolic PAP rather than on the systematic invasive hemodynamic assessment of PAPm and atrial pressure [5,25]. An echocardiographic sub-study of the TOPCAT trial showed that 36% of patients had a tricuspid regurgitant velocity of >2.9 m/s, corresponding to estimated systolic PAPs of at least 35 mmHg [33]. In the recent sub-analysis of the PARAGON-HF trial, the rate of PH-HFpEF was 31% [34]. Gerges et al. invasively assessed the prevalence of PH in HFpEF patients using the standard cut-off of PAPm > 25 mmHg, reporting a prevalence of 54.4% [8]. Access to precise PH-HFpEF statistics concerning the sub-classification of Ipc-PH and Cpc-PH is difficult to obtain; since a right heart hemodynamic assessment is not routinely performed in all patients with HFpEF, it makes it difficult to estimate the number of patients with increased vascular resistance and a progression of PH (Table 2). A retrospective analysis of 2587 PH-HFpEF patients showed a prevalence spanning from 8.8% to 3.5% using a diastolic pressure gradient (DPG) of >7 mm Hg or a PVR of >3 WU as the hemodynamic criteria. The report of Gergers et al., merging the retrospective and prospective data of nearly 4000 cardiac catheterizations, found a Cpc-PH rate of 22.6% for HFpEF using a DPG of >7 mmHg [16]. It is very important to identify and select subjects affected by CpcPH among PH-HfpEF patients because they have a substantially worse prognosis than those with IpcPH and may benefit from therapies that improve pulmonary vascular function [35].

## 4. Hemodynamic and Pathobiological Alterations in PH-HFpEF

PAH and PH-LHD are hemodynamically differentiated by pulmonary artery wedge pressure. In PAH, the mean PAWP is ≤15 mmHg, whereas in PH-LHD, the mPAWP is >15 mmHg [9]. This hemodynamic difference ensues from the chronic elevation in LV end-diastolic and left atrial pressure (LAP), which are triggers for the development of PH in HFpEF and HFrEF [36]. The elevation of pulmonary artery pressure (PAP) in the initial phase of the development of PH-HFpEF is, therefore, considered a passive reflection of LAP in the pulmonary circulation [37]. From a microscopic point of view, the main pulmonary vascular alteration occurs, initially, at the venous level, with increased vessel permeability, parietal thickness, and collagen deposition [5,12]. The barotrauma induces a process known as “alveolar-capillary stress failure”, or a reaction of the venules and capillaries, consisting of the disruption of endothelial permeability, alveolar flooding, and trigger of the extracellular matrix deposition [38]. In the first stages, the deposition of the extracellular matrix has the potential to absorb and accommodate fluid into the interstitium and is typical of isolated post-capillary PH [6]. If the stimulus is removed, this mechanism is reversible. However, chronic, elevated pressure induces the activation of metalloproteinases with true capillary remodelling, “arterialization” of pulmonary veins, and luminal narrowing [5,39]. Moreover, when LAP is long-standing increased, the process is reflected backward, causing arteriolar modifications with medial hypertrophy, intimal and adventitial fibrosis, and progressive luminal occlusion, but without the angioproliferative process seen in PAH [40] (Figure 1). When both venules and small pulmonary arteries are involved, an increased PVR and advancement toward combined pre- and post-capillary PH-HFpEF can occur. This phase of the disease is analogous to PAH endothelial dysfunction with capillaries and arteriole vasoconstriction, vascular obliteration [41], and pulmonary blood fluid redistribution from the basal to the apical district [4]. The initial alteration begins at the peripheral pulmonary vessels, but it is quickly transmitted to the medium and larger arterial lumen, up to the involvement of the two main branches of the pulmonary artery [4,11]. The common pathophysiological pathway is probably related to an imbalance between nitric oxide (NO) and endothelin-1, with soluble guanylate cyclase (sGC) inhibition [42,43]. In addition, hypoxia causes an imbalance in electrolytes transportation, resulting in altered alveolar fluid transport [44].

In CpcPH, the remodelling of the pulmonary arterial system contributes to the augmentation in PVR and to the decrease in pulmonary vascular compliance. Namely, in the pulmonary vascular bed, the arterial compliance (PAC) is distributed over the entire vessel system, and the distal vessels primarily determine the resistance and compliance, unlike the systemic circulation, where 80% of the compliance is from the proximal aorta [45]. As these two hemodynamic features represent the vascular load, their relationship has been analyzed. The studies report that their relationship is inverse, and the distribution is stable, as the pulmonary artery compliance (PAC) × PVR is constant [46,47,48,49]. The relation remains unaltered in PH, where the proportionality of pulmonary pressures is maintained. PAC can be considered as a surrogate of pulmonary arterial distensibility (pulsatile load) and calculated as stroke volume (SV)/ pulse pressure (PP), which doesn’t contain PAWP [49,50]. Therefore, PAC can be decreased as a consequence of PAWP’s increase, mediating changes in the PAPm at any given level of PAWP [5]. In the early phase of the development of PH, changes in PAC occur earlier than PVR elevation, and the PVR increase is very small and not detectable. Thus, PAC could be used as an early marker of the development of early pulmonary vessel disease and increased PVR [51]. Because of the unceasing elevation of PAWP in PH-HFpEF, the enhanced pulsatile and resistive load are reflected backward, thus affecting the right ventricle afterload. This strong connection between pulmonary vascular disease and the RV function is central to the development of PH, influencing prognosis and management.

## 5. Right Ventricle Mechanisms of Adaptations to Afterload: Ventriculoarterial Coupling and Uncoupling

In pulmonary hypertension, the increasing vascular load is transmitted to the right side of the heart, with right ventricle adaptation being a common consequence of both PAH and pulmonary hypertension in HFpEF (PH-HFpEF). The survival of patients with PAH and PH-HFpEF is closely related to right ventricular (RV) function [39,52,53,54,55,56]. Only recently have we learned how to understand the synchronous work of pulmonary circulation with the RV and the consequences of PH with regard to RV load and subsequent right ventricular dysfunction [49]. The increased afterload on RV leads to several adaptation mechanisms, such as hypertrophy and enhanced contractility. However, in the vast majority of patients, progressive dilatation and RV dysfunction occur [39,49]. An accurate evaluation of the right heart is, therefore, essential in the follow-up of patients with PH, either PAH or PH-HFpEF, as the assessment of RV adaptation and systolic dysfunction are pivotal for predicting the outcome in both conditions [57].

The augmentation of the resistive and pulsatile load of the pulmonary circulation affects RV function, causing the remodelling of the right ventricular chamber to counteract the elevated afterload. The progressive increase in PAP triggers a process based on the myocardial fibres’ adaptation and leads to concentric hypertrophy, with a higher mass-volume ratio and increased contractility in order to preserve systolic and diastolic function [39]. This reaction mechanism of the RV to afterload is called “ventriculoarterial coupling” (Figure 2). Mechanical coupling allows the RV to maintain the RV flow output at a minimal energy cost, and it depends on its ability to enhance contractility in response to increased pulsatile/resistive load [44,58,59]. The estimation of coupling can be obtained from the pressure–volume loops analysis. Ventricular contractility is given by the end-systolic pressure vs. end-systolic volume ratio, which gives the end-systolic elastance (Ees). Ventricular afterload is defined by arterial elastance (Ea), a measure of PVR (PVR × heart rate). The optimal coupling is assessed from the ratio between Ees and Ea, where the minimum energy cost corresponds to a ratio of 1 to 2. At this stage, there is an excellent balance between RV mechanical work and oxygen consumption, and an increase in contractility (Ees) of 4 to 5- fold can even take place [49]. The Ees/Ea ratio can be simplified with volumetric measures such as SV/ESV, which seems to deteriorate in CpcPH but preserved in IpcPH, therefore, conferring the information in the earlier stages of PH [44,49]. Monitoring the adaptational mechanisms of coupling in PH patients, including measures of SV, RV end-diastolic (RVEDV), and end-systolic volumes (RVESV), together with RV ejection fraction (RVEF), is important. Both RVEF and SV/RVESV are described as prognostic parameters in PH evolution. Advanced stages of the disease lead to a decrease in Ees due to ventricular remodelling and decompensation. The compensatory mechanism is imbalanced, with a reduction in stroke volume, dilatation, and uncoupling [49,60].

Ventriculoarterial uncoupling occurs when the stretching of myocardial fibres and chamber stiffness prevail over hypertrophy, thus unbalancing the relationship between cardiac contractility and vascular load. Maladaptive remodelling is represented by eccentric hypertrophy of the RV and subsequent systo-diastolic dysfunction [39]. In the advanced stages of the disease, the hypercontractile/hypertrophic state is not sufficient to preserve SV, the systolic function tends to decrease, and free wall hypokinesia occurs, leading to a progressive chamber dilatation. In order to maintain an energetically efficient situation to sustain CO, the heart rate increases and, given Ea as PVR × HR, Ea raises, decreasing the Ees/Ea ratio and, thus, leading to ventriculoarterial uncoupling [49]. RV excitation/contraction coupling starts impairing, with a loss of myocardial contractility reserve, an increased RV stiffness with a further elevation of intracardiac pressure, an affection of diastolic dysfunction, and an atrial emptying reduction [61]. Volumetric adaptation and dilatation occur in an attempt to counteract SV decrease. Thus, a follow-up of volumetric measures in PH patients, focusing on the PAPs and PAPm estimation that could reveal some fluctuations related to intrinsic hemodynamic conditions, pulmonary artery compliance, and vascular resistance, can identify the disease stage.

The left and the right ventricle uncoupling leads to augmented oxygen consumption, deterioration of ventricular interaction, and RV dyssynchrony. The left and right ventricle are actively interconnected, and they share both a diastolic phase and systolic phase. Ventricular interdependence can be altered because of both the lowering of the RV stroke volume and/or the interventricular septum bowing toward the left ventricle, which can hamper LV filling [62]. It has been recognised that, in the advanced stages of PH, the RV isovolumetric contraction is prolonged, while the LV is already in the diastolic phase [63]. The mechanical inefficiency is, therefore, related to the augmented RV’s post-systolic isovolumetric time, when the prolonged RV free wall contraction impairs the interventricular diastolic phase with right-to-left dyssynchrony. This explains why Doppler echocardiography dyssynchrony measures are related to prognosis [64]. Relaxation abnormalities, together with oxygen consumption inefficiency and reduced SV, lead to right heart failure, worsening PH prognosis and decreasing the patients’ survival.

## 6. Noninvasive Diagnostic and Prognostic Tools and Potential Applications

The impact of RV dysfunction and pulmonary hypertension was proven to be independently related to survival and outcome in left heart disease [39,65,66]. A multi-imaging approach, together with invasive right heart measures, is fundamental for RV evaluation, functional analysis, and pulmonary pressures assessment in order to define prognosis in HF. The gold standard technique for studying RV volumes and function is cardiac magnetic resonance because of its high temporal and spatial resolution. The difficult availability of this exam, together with its high costs, makes its use impossible on a large scale. Transthoracic echocardiography is the main technique recommended by the latest European guidelines to picture the effect of PH on dimensions, volumes, morphology, and kinetic of the right ventricle. They consider tricuspid regurgitation peak velocity, in addition to other echocardiographic findings, such as the estimation of systolic PAP, the IVC diameter, and the right ventricular outflow doppler measurements, which are indirect signs of pulmonary hypertension. Moreover, RV morphology may be evaluated by measuring the diastolic diameters at the basal or mid-ventricle level, below the tricuspid annular valve, and at the tricuspid chordal apparatus, respectively [1]. An estimation of PAP based on the Bernoulli equation has proven to be significantly related to the prognosis of HFpEF; its prognostic power increases with other functional and morphological parameters. Lam et al. proved that every 10 mmHg increase in PAPs is associated with a 1.2-fold increased risk of death, independent of age [67]. These data are confirmed by the Danish ECHOS study, where a PAP of >30 mmHg in HFpEF patients is associated with higher mortality [68]. Together with PAPs, tricuspid annular plane systolic excursion (TAPSE) is a marker of RV systolic function and an independent predictor of outcomes in patients with HF [69]. A threshold of 14 mm has a strong relation with prognosis [66,70].

The relation between TAPSE and PAPs was studied as an index of the RV length/force relationship (i.e., vetriculoarterial coupling) first by Guazzi et al. They demonstrated that a TAPSE/PAPs ratio < 0.36 mm/mmHg is associated with increased cardiac-related mortality in both HFpEF and HFrEF [71]. The same principle was related to hemodynamic measures by Gerges et al., thus validating the TAPSE/PAPs ratio of 0.31 mm/mmHg as the ideal cut-off for identifying CpcPH-HFpEF [8]. Another study by Ghio et al., involving 1663 patients with HFpEF and HFrEF, showed a strong relationship between PAPs and TAPSE in the first group. In this study, RV dysfunction was defined by reduced TAPSE; patients with HFpEF and pulmonary hypertension had markedly reduced TAPSE, while subjects with normal PAPs did not show a reduction in the tricuspid excursion. This revealed how there could be a strong dependency on the development of PH and subsequent RV dysfunction in HFpEF [66]. Accordingly, our group demonstrated a different RV diameter and TAPSE ratio in HfpEF and HFrEF, thus confirming the different adaptation of RV in two HF subtypes [72]. Together, these findings confirmed that RV dysfunction in HFpEF patients was strongly related to an enhanced afterload and PH.

Among other functional and morphological right ventricle measures, a pulsed wave Doppler applied on the right ventricular outflow tract can be useful to assess hemodynamic properties of PH. The presence of a mid-systolic notch was shown to be significantly related with RV dysfunction and CpcPH, whereas its absence was prevalent in IpcPH [35]. Moreover, an RVFAC of <35% and right free wall thickness were found to be related independently to increased risk of death, respectively, with a 2.4-fold and 1.4-fold [67,73].

New echocardiographic techniques, such as strain analysis and 2D/3D-speckle tracking echocardiography, should also be taken into consideration for the evaluation of the RV. A post-processing evaluation of the longitudinal functional analysis of the RV free wall, together with the characteristics we listed before, can add important information on right ventricle function [74].

The use of these parameters is of high value for the definition of the clinical and non-invasive probability of PH in HFpEF and for distinction from PAH. Each feature can have a prognostic and/or diagnostic value, but neither can be used alone to identify the presence of pulmonary hypertension. Together, with increasing interest in PH due to HFpEF, the difficulties of management, and the low availability of therapies, more studies are needed to assess the correct diagnostic and prognostic pathway in the field of pulmonary hypertension. A multi-modality imaging approach is fundamental for the definition and follow-up of RV coupling and uncoupling, with a focus on PH-LHD, as no therapies have been so far validated to improve its outcome and prognosis.

## 7. Conclusions

RV dilatation and dysfunction are well-known features of the adverse outcome of heart failure when LHD is complicated by pulmonary hypertension. RV dysfunction is often associated with advanced HF, systemic congestion, and worse clinical conditions. This scenario is related to increased left atrial pressures transmitted backward on pulmonary vessels, causing combined PH and right ventricle overload. Mechanisms of the development of PH in HFpEF appear to be significantly different compared to the haemodynamic adaptations typical of patients with HFrEF. They are mostly related to pulmonary vascular dysfunction, diffuse extracellular collagen deposition, and reduced pulmonary compliance in patients with suggestive echocardiographic findings and clinical phenotypes. When PH-HFpEF evolves from isolated post-capillary to combined pre- and post-capillary PH, hemodynamic changes overlap the typical alterations described in PAH. Specific studies investigating the triggers of PH in HFpEF are needed to precisely recognize the complex venous arterial and capillary alterations in order to achieve a better definition and a more appropriate targeted management.

## Figures and Tables

**Figure 1 ijms-23-04554-f001:**
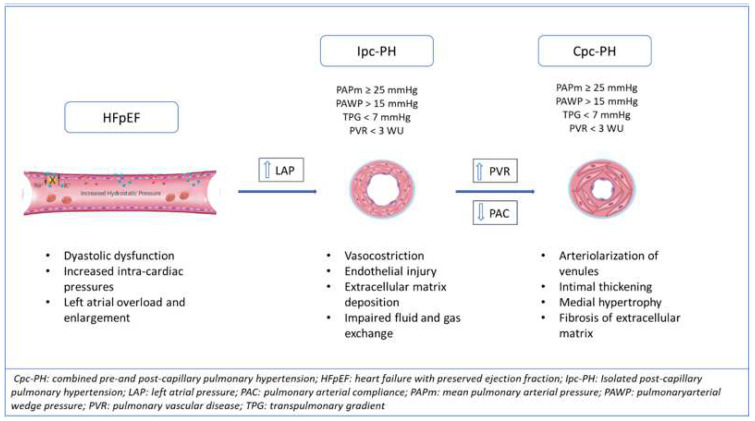
Development of pulmonary hypertension in HFpEF: pathobiological vessels’ alterations and hemodynamic changes from isolated pre-capillary (Ipc) to combined pre- and post-capillary hypertension (Cpc-PH).

**Figure 2 ijms-23-04554-f002:**
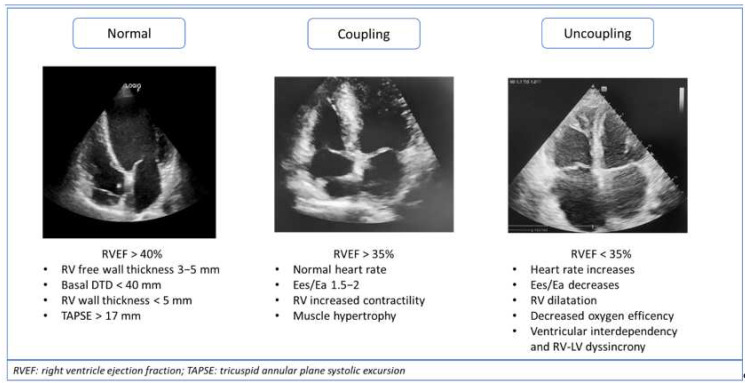
Mechanisms of RV adaptation in pulmonary hypertension (PH) progression related to heart failure with preserved ejection fraction (HFpEF): ventriculo-arterial coupling and uncoupling.

**Table 1 ijms-23-04554-t001:** Clinical, echocardiographic, and hemodynamic differentiation of pulmonary arterial hypertension and pulmonary hypertension in left heart disease.

Parameters		
PAH	Clinical	PH-LHD
<65	Age	>65
Rare	Hypertension	Common
Uncommon	Metabolic syndrome	Common
Uncommon	Diabetes	Common
Uncommon	Obesity	Common
Uncommon	CAD	Common
Rare	Artial Fibrillation	Common
Rare	Obstructive sleep apnoea	Common
**Echocardiographical**		
Usually small	LA size	Enlarged
Increased	RA/LA ratio	Normal
Typical	RV hypertrophy	Atypical
Atypical	LV Hypertrophy	Typical
<8	Lateral E/e’	>10
<1	E/A ratio	>1
Common	RVOT notching	Rare

CAD: coronary artery disease; DPG: diastolic pulmonary gradient; LA: left atrium; LAP: left atrial pressure; LV: left ventricle; PAPm: mean pulmonary arterial pressure; PAWP: pulmonary arterial wedge pressure; PVR: pulmonary vascular resistance; RA: right atrium; RV: right ventricle; RVOT: right ventricle outflow tract; Pulmonary Arterial Hypertension: PAPm ≥ 25 mmHg; PAWP < 15 mmHg; PVR ≥ 3 WU; DPG ≥ 7 mmHg; LAP = 0–5 mmHg. Pulmonary Hypertension in Left Heart Disease: PAPm ≥ 25 mmHg; PAWP ≥ 15 mmHg; PVR < 3 WU; DPG < 7 mmHg; LAP > 8 mmHg.

**Table 2 ijms-23-04554-t002:** List of HFpEF trials with hemodynamic evaluation and invasive heart catheterization.

Study	N° Subjects	Hemodynamic Characteristics	Duration
Guazzi	44	mPAP > 25 mmHg PCWP > 15 mmHg TPG > 12 mmHg	1 year
Hoendermis	52	mPAP > 25 mmHg PCWP > 15 mmHg DPG ≥ 17 mmHg	12 weeks
DILATE 1	39	mPAP > 25 mmHg PCWP > 15 mmHg	12 weeks
BADDHY	20	mPAP > 25 mmHg PCWP > 15 mmHg 6 MW distance > 150 mt and <400 mt	12 weeks
DYNAMIC	114	mPAP > 25 mmHg PCWP > 15 mmHg	26 weeks
PASSION	320	PAWP > 15 mmHg mPAP > 25 mmHg PVR > 3WU	NA
SERENADE	300	PCWP or LVEDP > 15 mmHg	-

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
