# Peer review of "Structural and Hemodynamic Changes of the Right Ventricle in PH-HFpEF"

_ijms, 2022, doi:10.3390/ijms23094554_

Round 1
Reviewer 1 Report
This review describes the right ventricular consequences of combined Pulmonary Hypertension and Heart Failure with Preserved Ejection Fraction (HFpEF). The authors present a comprehensive overview of key features of right ventricular dysfunction in a well-structured and, for the most part, well-written manuscript. I only have minor comments, primarily pertaining to some typographical errors.
- Throughout the manuscript, you write “et Al”, please change to “et al.,”.
- Line 12: Change “under” to “in”
- Line 39: Change “on” to “in”.
- Line 46: Change “increase” to “increases”..
- Line 92: Change “study group” to “studies”.
- Line 177-119: This doesn’t make sense as currently written. Maybe, “…; the reported prevalence among the HFpEF population ranges from 18% to 83%, depending on the group studied.”?
- Line 149: Change “reflexion” to “reflection”.
- Line 154: Remove both occurrences of, “of”.
- Line 159: “the/a” ?
- Line 248: remove “vascular”.
- Line 251: change the start of the sentence to, “The left and right ventricles…”
- Line 254: Change “septal” to “septum” and add “which” before can.
- Line 267-269: Rewrite sentence starting, “Thus,”. It does not make sense as currently written.
Figures and Tables:
- Table 1 – To make this review more comprehensive, you could add in citations to show where the data reporting the incidence of the clinical or echo derived parameters comes from. This would make this manuscript much stronger.
- Figure 1 – This a great figure. Why is “Arterialization” surrounded by marks?
- Figure 2 – I am unsure whether these echo screenshots really help tell the story. They are foreshortened and have the lateral walls clipped. If you decide to include them, use images that are on the same scale, so that it is easier to show any easily discernible differences in morphology.
Author Response
We are grateful for your comments and revision. We applied all the modifications you suggested inside the text. The references linked to data in table 1 have been added and figure 2 has been modifided to improve the utility of images concerning the evolution of the right ventricle in ventriculoarterial coupling.
Reviewer 2 Report
This is overall a nicely written review.
I have no comments
Author Response
We are glad you liked our paper and we appreciate your nice comment.
Reviewer 3 Report
This manuscript presents the review of structural and hemodynamic changes of the right ventricle in PH-HFpEF.
The authors provided various data of PH-HFpEF that is characterized by specifical demographic, clinical and echocardiographic features compared to PAH phenotype. In this review, the pathological features of the older population, female gender, including hypertension, higher rate of atrial fibrillation, and metabolic syndrome were used.
The review article is important to understand that complex venous arterial and capillary alterations should be considered for an investigation of the triggers of PH in HFpEF that will allow for a better understanding of the mechanism of heart failure and could be considered for publishing in the International Journal of Molecular Sciences.
However, some minor corrections should be made to the manuscript such as a correction of text font.
Author Response
We are grateful for your comments to our review. Small modifications have been applied to text and figures. In particular, text form is now even thorough the whole document.